# A resilience group training program for people with multiple sclerosis: Results of a pilot single-blind randomized controlled trial and nested qualitative study

**Ambra Mara Giovannetti**[1,2,3]*, **Rui Quintas**[3,4], **Irene Tramacere**[5], **Andrea Giordano**[1,6], **Paolo Confalonieri**[3], **Michele Messmer Uccelli**[7], **Alessandra Solari**[1], **Kenneth Ian Pakenham**[2]

**1** Unit of Neuroepidemiology, Fondazione IRCCS Istituto Neurologico Carlo Besta, Milan, Italy, **2** School of Psychology, Faculty of Health and Behavioural Sciences, University of Queensland, Brisbane, QLD, Australia, **3** Unit of Neuroimmunology and Neuromuscular Diseases, Fondazione IRCCS Istituto Neurologico Carlo Besta, Milan, Italy, **4** Clinical and Experimental Epileptology Unit, Fondazione IRCCS Istituto Neurologico Carlo Besta, Milan, Italy, **5** Department of Research and Clinical Development, Scientific Directorate, Fondazione IRCCS Istituto Neurologico Carlo Besta, Milan, Italy, **6** Department of Psychology, University of Turin, Turin, Italy, **7** Italian Multiple Sclerosis Society and Research Foundation (FISM), Genoa, Italy

* ambra.giovannetti@istituto-besta.it

**Data Availability Statement:** All relevant data are within the paper and its Supporting Information files.

## Abstract

### Introduction

An Australian case series study demonstrated the effectiveness of the REsilience and Activities for every DaY for people with multiple sclerosis (READY for MS), a resilience group training program based on Acceptance and Commitment Therapy, in improving quality of life in people with MS. This study aimed to evaluate the feasibility and acceptability of the Italian READY for MS program, and to preliminary assess its efficacy when compared to an active control intervention (group relaxation).

### Methods

Single-blind phase II randomized controlled trial (RCT) and nested qualitative study (ISRCTN registration number: 38971970). Health-related quality of life (primary study outcome), mood, resilience, psychological flexibility and its protective factors were measured at baseline, after seven, 12 and 24 weeks. READY participants completed the purpose-built satisfaction questionnaire after 12 weeks. After trial completion, the control group also received READY. One-to-one participant interviews were conducted within three months of finishing the READY groups.

### Results

Four intervention groups were conducted with 39 participants (20 READY, 19 relaxation). Two patients (READY) withdrew before beginning the intervention due to unexpected work commitments. Feasibility and acceptability of READY were good, with high participant

**Funding:** Supported by FISM - Fondazione Italiana Sclerosi Multipla – cod. 2016/B/3 and financed or co-financed with the '5 per mille' public funding. The funding source had no role in study design, data collection, data analysis, data interpretation or report writing.

**Competing interests:** AS reports grants from Fondazione Italiana Sclerosi Multipla (FISM), during the conduct of the study; personal fees from Biogen Idec, Merck Serono, Novartis, Almirall, and Excemed. As a staff member of the University of Queensland and co-Author of the READY program, KP receives royalties from UniQuest for commercial (not research) licensing arrangements entered into by third parties who want to deliver the program. PC acted as an Advisory Board member of Biogen and Novartis; received funding for traveling from Biogen, Merck, Teva, Novartis; received honoraria for speaking or writing from Biogen and Novartis. He received support for research project by Novartis and Merk and is involved as principal investigator or co-investigator in clinical trials for Teva, Novartis, Biogen and Merk. RQ received funding for conducting research and clinical projects from AIM Education (Novartis) and Fondazione Serono (Merck) in 2019. This does not alter our adherence to PLOS ONE policies on sharing data.

engagement and satisfaction. No statistical effects of READY were detected vs relaxation. Thirty participants were interviewed (18 READY; 12 relaxation + READY). Content data analysis revealed seven overarching themes: "Attitudes towards participation"; "Perceptions of program composition"; "Program impacts on life domains"; "Program active elements"; "Program improvement trajectories"; "Program differences and similarities"; "Suggested READY improvements".

## Conclusion

READY was well accepted by MS patients with varied socio-demographic and clinical characteristics. Qualitative (but not quantitative) data provided evidence in favour of READY. Our findings will inform methodological and intervention refinements for the multi-centre RCT that will follow.

## Introduction

Multiple sclerosis (MS) is an inflammatory, progressive demyelinating disease of the central nervous system. With a lifetime risk of 1 in 400, it is the most common cause of progressive neurological disability in young adults [1]. The most recent study on the global prevalence of MS estimated that approximately 2.3 million people worldwide have MS, with Canada, USA and some European countries, including Italy, having the highest prevalence rates [2]. The Italian MS Foundation estimated that there are more than 122,000 persons with MS (PwMS) in Italy [3]. Given that MS typically manifests in young adulthood [4], the impact of diagnosis is particularly distressing as it has the potential to significantly interfere with life goals [5], and negatively impact on work-related activities [6]. Patients' experience of MS is characterized by remissions, relapses, possible persistent disability and continuous progression. As a result, PwMS often have to deal with uncertainty about disease progression, loss of function, changes in life roles and a variety of symptoms [7]. For these reasons, adjusting to MS can be highly demanding [8], and the disease can be a consistent source of stress. In fact, PwMS have poorer quality of life (QoL) than healthy controls and people with other chronic diseases; lifetime prevalence is approximately 50% for depressive symptoms and 35% for anxiety disorders [9, 10]. In addition, evidence suggests an association between psychological stress and subsequent relapses in MS, with the occurrence of stressful life events purported to lead to a greater risk for relapses [11].

Resilience is an internal resource for alleviating the adverse effects of stress and sustaining good mental health through adversity [12]. It entails the process of negotiating, managing and adapting to significant stressors or trauma through drawing on internal (i.e. mindfulness, acceptance, cognitive flexibility and active coping), and external (i.e. social support, financial capital and community services) resources [13]. In times of adversity, people with low resilience have a higher risk of experiencing poor QoL, emotional burden and interpersonal difficulties [14]. Moreover, they can adopt health compromising behaviors and experience somatic complaints and poor physical health [15]. Prolonged stress together with poor psychosocial functioning may negatively affect physical health through different mechanisms, such as hypertension and blood pressure reactivity to stress, pro-inflammatory cytokines and the development of metabolic syndrome [16]. Given that PwMS have lower resilience than community samples and people with other chronic illnesses [17], they are particularly vulnerable

to low QoL and well-being. Therefore, targeted interventions aimed at fostering resilience are crucial in helping PwMS deal with their illness-related stressors and improve their QoL.

Leppin et al's [12] meta-analysis of resilience training programs in adults showed modest but consistent benefits in improving a number of mental health outcomes. A more recent systematic review demonstrated that resilience training promotes a range of positive psychosocial outcomes in people with chronic illnesses [18]. Resilience training programs have been shown to improve QoL, anxiety, depression, perceived stress and well-being in adults with cancer [19, 20, 21], congenital heart disease [22], diabetes [23], and neurofibromatosis [24]. Despite these promising findings, a recent narrative review of resilience training identified numerous methodological limitations. First, evidence of the benefits of resilience training remains limited; second, resilience training interventions are often not well differentiated from other forms of training; third, the effects of the training on psychological functioning are strongly influenced by the outcome measures selected and the setting of the training [25].

In recent years, an Australian team developed and tested an Acceptance and Commitment Therapy (ACT)-based group resilience-training program: the REsilience and Activities for every DaY (READY). The READY program was initially trialed in a workplace setting [26, 27] and then adapted and successfully applied to different health conditions: cancer [28] diabetes [29] and MS [30]. Findings showed READY for MS had beneficial impacts on resilience, health related QoL, depression, stress and protective factors (managing difficult thoughts, values and acceptance) in PwMS [30]. READY for MS is based on ACT which is a recent variant of cognitive behavior therapy. ACT is informed by the psychological flexibility framework. Psychological flexibility is defined as *"the ability to contact the present moment more fully as a conscious human being, and to change or persist in behavior when doing so serves valued ends"* [31]. It is fostered by six ACT processes: (1) acceptance–openness to experience, (2) cognitive defusion–observing thoughts rather than taking them literally, (3) present moment awareness–mindfulness, (4) self as context–contact with a sense of self that is continuous and provides flexible perspective taking, (5) values–freely chosen personally meaningful life directions, (6) committed action–values-guided effective action [31]. Each process has been shown to be related to better mental health, lower risk of disease, better health outcomes for those already diagnosed with illness [29, 31–34]. Psychological flexibility plays a key role in promoting resilience [35]. Furthermore, psychological flexibility processes have been shown to mediate the beneficial effects of resilience training in PwMS [30].

In view of the promising preliminary pilot data on the READY for MS resilience training intervention and taking into consideration the limitations highlighted by Forbes and Fikretoglu [25], the present study aimed to evaluate the program using greater methodological rigor based on the Medical Research Council (MRC) framework for developing and evaluating complex interventions. The MRC framework has a phased approach, from a pre-clinical research phase to a final phase in which the intervention is introduced into the health service, leading to a theory-driven intervention: a "bottom up" development which is crucial to design a phase III trial with an appropriate theory and pilot work [36]. Further, both quantitative and qualitative methods are used and integrated within the framework, in order to better appraise the effects of the (complex) intervention as a whole and its components. For this reason, the project evaluating the Italian version of the READY for MS program is composed of two phases: 1) a pilot randomized controlled trial (RCT) with a nested qualitative study; 2) a multi-centre phase III RCT, with an ancillary study on the impact of the training on the psychologists who will learn and deliver the READY for MS program. The aim of the present paper is to present the results of the pilot RCT (phase 1), which evaluates the feasibility and acceptability of the Italian READY for MS program and its efficacy by comparing it to an active control intervention (group relaxation).

## Material and methods

This is a pilot, single blind RCT with a nested qualitative study comparing Italian READY for MS (hereinafter called READY) with relaxation. The consolidated criteria for reporting a pilot or feasibility trial (CONSORT; S1 Appendix) [37] and qualitative research (COREQ; S2 Appendix) [38, 39] guided the presentation of findings. Data were collected via questionnaires (see "Measures" below) immediately before the intervention (baseline T0), after the intervention (seven weeks after baseline T1), after the booster session (12 weeks after baseline T2) and at three months follow-up (24 weeks after baseline T3). In addition, the examining clinician (not involved in intervention delivery and blinded to patient assignment) administered the Schedule for the Evaluation of the Individual Quality of Life-Direct Weighting (SEIQoL-DW) [40] at T0, T2 and T3. The SEIQoL-DW was administered in a dedicated room of the hospital. The examining clinician (RQ) was a researcher and clinical psychologist working at the MS Centre of the Fondazione IRCCS Istituto Neurologico Carlo Besta (Milan), with expertise in qualitative methodologies and specifically trained to administer the SEIQoL-DW interview.

The following clinical information was reported on the case reports form by the patient's referring neurologist: expanded disability status sale (EDSS) [41] score; MS course (relapsing remitting, primary progressive, secondary progressive), and ongoing treatment at T0. At the end of the three months follow-up, participants allocated to relaxation were offered READY. READY participants completed the purpose-built Participant Satisfaction Questionnaire at T2. Participants assigned to the READY condition and control group participants who later completed READY were interviewed to obtain additional intervention feedback. Additional weekly-administered process measures assessed participants' attendance and homework completion, and facilitator perspectives. The study was conducted at the MS centre of the Fondazione IRCCS Istituto Neurologico Carlo Besta, Milan, Italy. It was carried out in accordance with the recommendations of the Declaration of Helsinki. The protocol (S3 Appendix; DOI: dx.doi.org/10.17504/protocols.io.bcqjivun) received ethical clearance from the ethics committees of the Fondazione IRCCS Istituto Neurologico Carlo Besta (8 February 2017, internal ref: 37; amendment approved 06 September 2017, internal ref: 43) and The University of Queensland (8 May 2018, clearance number: 2018000942/Review 08022017). All participants gave written informed consent. The study was registered on the ISRCTN registry (ISRCTN registration number: 38971970), 5 June 2018, before enrolment completion. The authors confirm that all ongoing and related trials for this intervention are registered.

### Study participants and recruitment

**Participant eligibility.** People were included in the study if they met all of the following criteria: diagnosis of MS [42], age ≥18 years; signed informed consent, resilience score <83, able to attend intervention group sessions, and fluent Italian speaker. They were excluded from the study if one or more of the following criteria were met: severe cognitive compromise (Mini Mental State Examination <19), ongoing psychotherapy in the preceding six months, ongoing practice in meditation or other mind-body therapies, major psychiatric disorders (including psychotic disorders or active substance abuse problems), pregnancy, MS diagnosis for less than three months, and one or more relapses in the last month.

*Recruitment and trial procedures.* A flyer, which included a general overview of the study and contact details, was disseminated via e-mail to the MS Centre patients by the MS Centre team. People who showed interest in participating in the study were contacted by the study coordinator. Subsequently, the examining clinician made an appointment with those patients who met the inclusion criteria and agreed to participate in the study, checked all eligibility criteria and performed the baseline evaluation (T0) in a dedicated room of the hospital.

Information on all screened people and reasons for exclusion were recorded. Participants were then assigned to the READY or control condition. Randomization was provided by an independent randomization service at the Neuroepidemiology Unit, using computer-based stratified randomization (2 factors: Expanded Disability Status Scale (EDSS) [41] score < 2.0 and ≥ 2.0, and CD-RISC 25 score < 50 and ≥ 50). Participants were allocated to READY or relaxation program in a 1:1 ratio. Confirmation e-mails were sent to the study coordinator. The interventions started within two weeks of the baseline assessment.

## Interventions

**READY.** READY is an adult ACT informed group resilience training program which consists of seven weekly 2.5 hour sessions plus a 2.5 hour 'booster' session approximately five weeks after the seventh session. Content of the seven weekly sessions is as follows: an introductory module (Introduction to the READY Resilience Model), five modules focusing on each of the six ACT processes (Mindfulness, Acceptance, Cognitive Defusion, Self-as-Context, Values and Meaningful Action), and a review module (Review and Future Planning). The booster session provides a review of the program content. The program has a facilitator manual, participant workbook, and audio recordings of mindfulness exercises. Throughout the program, participants are encouraged to share their progress and experience of applying the READY strategies and techniques. It incorporates a blend of psychoeducation and experiential exercises, combined with readings and homework exercises that participants are encouraged to practice between sessions [30]. The study coordinator (AMG) conducted all sessions.

**Control intervention.** The control condition consisted of a group relaxation program based on autogenic training. The intervention consisted of seven, 1-hour weekly group sessions, followed by a 'booster' session after five weeks. This control program matched the study intervention in number of sessions and schedule (but not in session content and length) in order to control for the non-specific effects of READY. The program had a facilitator manual, participant workbook, and audio recordings of relaxation exercises. The study coordinator (AMG) conducted all sessions.

## Measures

We used the Italian versions of the MSQOL-54 [43], the Hospital Anxiety and Depression Scale–HADS [44], the 10-item Perceived Stress Scale—PSS (courtesy of Fossati), The Mindful Attention Awareness Scale—MAAS [45], the 20-item Valued Living Questionnaire—VLQ [46], and the Acceptance and Action Questionnaire II—AAQ-II [47]. For the Connor-Davidson Resilience Scale—CD-RISC 25 we used the unpublished Italian version (courtesy of Davidson). The Comprehensive assessment of ACT processes—CompACT and the Drexel Defusion Scale–DDS, were translated into Italian for the study (papers in preparation). These patient reported outcome measures were administered in the order presented in the following section.

**Primary outcome measure.** The MSQOL-54 is a health-related QoL measure that comprises the generic Short-Form 36-item (SF-36), plus 18 MS-specific items [48]. The 54 items are organized into 12 multi-item and two single item subscales. As for the SF-36, two composite scores (Physical Health Composite, PHC, and Mental Health Composite, MHC) are derived by combining scores of the relevant subscales. The MSQOL-54 has well documented validity in terms of content, constructs, reliability [48], discrimination and responsiveness [49]. To limit multiple comparisons, we primarily assessed changes in the MHC.

Secondary outcome measures. *Mood*. The HADS is a well-validated measure that consists of two seven-item subscales to assess anxiety and depressive levels. Higher scores indicate

higher levels of depressive or anxiety symptoms [50]. Unlike many similar measures, the HADS excludes somatic symptoms of anxiety and depression, which may overlap with physical illness [50].

The PSS assesses the extent to which life situations are appraised as stressful. Higher scores indicate higher perceived stress [51].

*Resilience*. The CD-RISC 25 was used to assess psychological resilience. It is composed of 25 items, each rated on a 5-point scale (0–4), with higher scores reflecting greater resilience. The scale demonstrated good psychometric properties [52].

*General measure of ACT processes*. The 23-item CompACT has three subscales: openness to experience (OE), behavioral awareness (BA), valued action (VA). A total score is calculated by summing the three subscale scores. Items are rated on a 7-point Likert scale. The full-scale CompACT total score ranges from 0–138, with higher scores indicating greater psychological flexibility. The CompACT demonstrated good internal consistency, and converged and diverged in theory-consistent ways with other measured variables: higher levels of psychological inflexibility were associated with higher distress and lower health and well-being [53].

*Mindfulness*. The MAAS is a 15-item scale which assesses dispositional mindfulness across interpersonal, cognitive, physical, emotional, and general domains. Items are rated on a 6-point Likert scale, and responses are then summed with higher scores indicating greater mindfulness. The MAAS has demonstrated adequate validity, internal reliability and sensitivity to change [54].

*Values and meaningful action*. The VLQ measures the relative importance of certain life domains and the consistency of behaviors corresponding with identified personal values. Respondents are asked to rate the 10 life domains on a 1–10 scale on level of importance (importance subscale) and how consistently they have lived in accord with those values in the past week (consistency subscale). Higher scores indicate greater importance and consistency. The VLQ displays good inter-item consistency, test-retest reliability, and construct validity [55].

*Acceptance*. The AAQ-II is a 10-item self-report measure of acceptance and experiential avoidance. Items are rated on a 7-point Likert scale. Higher scores indicate lower acceptance and lower scores reflect greater acceptance. It has been shown to have good internal reliability and convergent validity [56].

*Cognitive defusion*. The DDS measures psychological distance from thoughts and feelings. Respondents are asked to read a definition of defusion prior to indicating the extent to which they would normally be in a state of defusion across ten different scenarios, using a 6-point Likert scale. Higher scores indicate greater ability to defuse from distressing thoughts and feelings [57].

*Individualized QoL*. The SEIQoL-DW is an interview-based instrument to assess the level of functioning and relative importance of areas of life identified by the respondent. The evaluation is based on three steps: (a) identifying the 5 most important QoL areas; (b) rating the relative importance of each identified area by the use of a pie chart whereby the size of its section can be adjusted to reflect the relative importance of each area (displays a 0–100 scale); (c) assigning a satisfaction score to each of the five areas. The SEIQoL-DW index is calculated by multiplying the importance of each identified area by the related weight, and summing the values for each area. The index score can range from zero (worst possible) to 100 (best possible) [40].

*Participant satisfaction questionnaire*. A purpose-build questionnaire was used to obtain participant feedback on the READY program. It had four sections: usefulness of the READY program in promoting the six ACT processes (six items), global evaluations of the READY

program (five items), and satisfaction with the READY Personal Plan (five items), and ratings of commitment to the READY Personal Plan after each session.

## Nested qualitative study

Two interview guides (S4 Appendix) were created: interview guide A for participants allocated to READY, and interview guide B for those assigned to relaxation and who then received READY. The interview guides had open-ended questions and prompts designed to elicit participants' accounts of their experience of the intervention(s). Interviews lasted a maximum of one hour and were audio-recorded and transcribed verbatim. No changes were needed in the interview guides after piloting them.

**Qualitative study participants and recruitment.** Participants were informed about the possibility of participating in an individual interview when recruited. They were invited to participate in the qualitative study by e-mail or telephone within three months of completing READY. They were then fully informed of the aims and requirements of this action and provided informed consent. All trial participants who received READY were invited to participate. The examining clinician conducted the interviews in a dedicated room of the hospital or by phone (when a participant was unable to visit the hospital).

## Analyses

Quantitative analysis. *Sample size calculation.* A sample size of 32 patients (18 per arm) achieves 80% power to detect a difference of 14 (standard deviation 22) in the MSQOL-54 MHC in a design with three repeated measurements, assuming an intra-subject correlation between observations of 0.75, an alpha error of 0.05, and 20% of the patients lost to follow-up (S5 Appendix). Because no data on the READY for MS were available when the study was designed, our estimate was based on the large effect size for health-related QoL ($d$ = 0.80) assessed by the Profile of Health-related Quality of Life in Chronic Disorders scale [58] in a RCT of a group mindfulness intervention for PwMS [59], and on available data on the MSQOL-54 MHC [49, 60].

*Statistics.* The normality assumption was tested with the Shapiro-Wilk test. Between-group comparisons were performed using either the two-sided unpaired t-test or the Wilcoxon two sided two-sample test for continuous variables depending on data distribution, and the chi-squared test for categorical variables. Correlations were computed using Spearman's or Pearson's coefficients depending on data distribution. Longitudinal changes were analyzed using repeated measures mixed-effects models with identity covariance structure. Group effect was included if the intraclass correlation coefficient (ICC) was > 0.20. All tests were two-tailed, and values of $p < 0.05$ were considered significant. All data were analyzed according to the intention-to-treat principle (ITT). A per protocol (PP) analysis was also performed. Statistical analyses were performed using Stata Statistical Software version 15 (StataCorp. 2017. Stata Statistical Software: Release 15. College Station, TX: StataCorp LLC).

**Qualitative analysis.** Content analysis was used to code the interview data and to identify themes that captured key concepts and processes. The analysis was performed by two psychologists (AMG; RQ). After the first three interviews were transcribed, the analysis proceeded iteratively, interview by interview. This method allows early insights to be explored more fully in later interviews and interview guides to be modified if necessary [61]. Analysis was inductive and involved line-by-line coding with codes and categories derived from narratives. A two-step coding scheme was applied. The first level codes came from sentences used directly by participants. This allowed critical and analytical examination of the data, generation of new ideas and indications for further data collection. A second step was used to aggregate data and to further refine the emerging codes and categories.

## Results

### RCT

Four intervention groups were conducted (two READY and two relaxation groups) between April 2017 and January 2018 (first recruitment start date 16 March 2017, end date 30 March 2017; follow-up completed in September 2017. Second recruitment start date 9 October 2017, end date 23 October 2017; follow-up completed in April 2018). The rate of recruitment was high (70%) and the most common reasons for refusal were time conflicts with work. Thirty-nine patients participated: 20 were assigned to READY, 19 to relaxation. Two participants (READY) withdrew before beginning the intervention due to unexpected work commitments (See Fig 1).

Most participants (total sample = 97.3%; READY = 94.4%; Relaxation = 100%) attended more than half the program, and more than three quarters (total sample = 75.7%; READY = 77.7%; Relaxation = 73.7%) attended at least seven out of the eight sessions. Only one participant attended 50% of the sessions (READY). The most common reasons given for missing sessions were time conflicts with work/study. No differences were found between ITT and per protocol results, hence only the ITT analyses are presented below. Baseline demographic and clinical characteristics of the sample are presented in Table 1.

No statistically significant differences were detected between READY and relaxation at baseline on any variables except for a higher MSQOL-54 MHC mean score in the READY arm ($p < 0.05$). Internal reliability coefficients were $\geq 0.70$ for all outcome measures. The group effect was not included in the model as the ICCs (overall and within each intervention) were $< 0.07$.

Repeated measure analyses revealed no significant effect of READY over relaxation on the MSQOL-54 MHC and the other outcome measures (Table 2). A significant ($p < 0.02$) effect was found for time in health-related QoL, resilience, anxiety, depression, perceived stress, psychological flexibility (particularly openness to experience and behavioral awareness), valued living (VLQ total score and consistency), cognitive defusion and acceptance (Table 2).

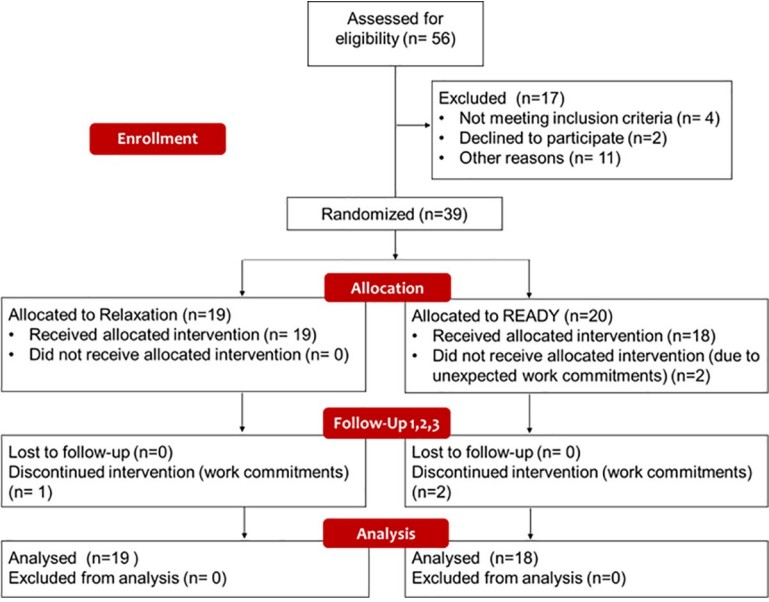

**Fig 1. Italian READY for MS pilot RCT CONSORT flowchart.**

**Table 1. Baseline demographic and clinical characteristics.** EDSS = Expanded Disability Status Scale; MS = multiple sclerosis.

| | Total (N = 37) | READY (N = 18) | Relaxation (N = 19) |
|---|---|---|---|
| Characteristics | | | |
| Age (years)–mean (SD) | 45.7 (9.1) | 44.8 (10.1) | 46.53 (8.3) |
| Women–n (%) | 22 (59) | 13 (72) | 9 (47) |
| Education–n (%) | | | |
| Middle school diploma | 9 (24.3) | 2 (11.1) | 7 (36.8) |
| High school diploma | 8 (21.6) | 3 (16.7) | 5 (26.3) |
| Degree | 16 (43.2) | 9 (50.0) | 6 (36.8) |
| PhD | 4 (10.8) | 4 (22.2) | 0 (0.0) |
| Marital status–n (%) | | | |
| Single | 14 (37.8) | 8 (44.5) | 6 (31.6) |
| Married | 21 (56.8) | 9 (50.0) | 12 (63.1) |
| Divorced | 2 (5.4) | 1 (5.5) | 1 (5.3) |
| Employment status–n (%) | | | |
| Full time employed | 21 (56.8) | 11 (61.1) | 10 (52.6) |
| Part-time employed | 2 (5.4) | 0 (0.0) | 2 (10.5) |
| Freelance | 6 (16.2) | 2 (11.1) | 4 (21.1) |
| Student | 2 (5.4) | 2 (11.1) | 0 (0.0) |
| Unemployed | 2 (5.4) | 1 (5.6) | 1 (5.3) |
| Retired | 4 (10.8) | 2 (11.1) | 2 (10.5) |
| Autonomy–n (%) | | | |
| Independent | 31 (83.8) | 14 (77.8) | 17 (89.5) |
| Partial assistance | 5 (13.5) | 3 (16.7) | 2 (10.5) |
| Total assistance | 1 (2.7) | 1 (5.5) | 0 (0.0) |
| MS type–n (%) | | | |
| Relapsing remitting | 30 (81) | 14 (78) | 16 (84) |
| Secondary progressive | 6 (16) | 3 (17) | 3 (16) |
| Primary progressive | 1 (3) | 1 (6) | 0 (0) |
| Disease duration (years) | 12.2 (10.7) | 13.7 (12.4) | 10.7 (8.9) |
| EDSS score–Median (min-max) | 2 (0–6.5) | 2 (0–6.5) | 2 (0–6.5) |
| Disease Modifying Treatment–n (%) | | | |
| None | 4 (10.8) | 2 (11.1) | 2 (10.5) |
| Interferon beta 1a | 7 (18.9) | 3 (16.7) | 4 (21.1) |
| Glatiramer acetate | 2 (5.4) | 1 (5.6) | 1 (5.3) |
| Dimethyl fumarate | 12 (32.4) | 7 (38.9) | 5 (26.3) |
| Teriflunomide | 2 (5.4) | 1 (5.6) | 1 (5.3) |
| Fingolimod | 8 (21.6) | 2 (11.1) | 6 (31.6) |
| Natalizumab | 1 (2.7) | 1 (5.6) | 0 (0.0) |
| Alemtuzumab | 1 (2.7) | 1 (5.6) | 0 (0.0) |

Fig 2 shows trend lines for the primary and selected secondary outcome variables (graphs for all outcomes are presented in S6 Appendix). The improvement trends were similar in the two arms for both the MSQOL-54 MHC (Fig 2A) and CompACT TOT (Fig 2C). However, READY participants appeared to have higher baseline scores on both measures. With regards to MSQOL-MHC, 8/37 (21.6%) had a MSQOL-54 score higher than 70: 33.3% (6/18) in the READY arm; 10.5% (2/19) in the relaxation arm. Figures for the CD-RISC 25 (Fig 2B), CompACT Openness to Experience subscale (Fig 2D) and DDS (Fig 2G) showed that the READY arm improved more than relaxation, and with the exception of DDS this difference increased

**Table 2. Repeated measures analysis (intention-to-treat).**

|  | α | READY (18) | | | | Relaxation (19) | | | | P value | | |
|---|---|---|---|---|---|---|---|---|---|---|---|---|
|  |  | T0 | T1 | T2 | T3 | T0 | T1 | T2 | T3 | T | R | TxR |
| MSQOL-54 MHC | 0.8 | 64.0 (4.0) | 70.7 (4.2) | 72.5 (3.9) | 73.2 (4.2) | 52.8 (3.8) | 58.2 (4.1) | 63.2 (3.8) | 62.7 (4.1)§ | <0.01 | 0.02 | 0.87 |
| MSQOL-54 PHC | 0.7 | 63.1 (3.5) | 65.1 (4.1) | 64.8 (4.2) | 67.3 (4.4) | 56.3 (3.4) | 59.0 (4.0) | 63.6 (4.1) | 62.7 (4.3)* | <0.01 | 0.39 | 0.22 |
| HADS-D | 0.7 | 5.3 (0.8) | 3.6 (0.9) | 3.5 (0.8) | 3.4 (0.9)* | 6.7 (0.8) | 6.5 (0.9) | 5.7 (0.8) | 5.5 (0.9) | <0.01 | 0.05 | 0.50 |
| HADS-A | 0.9 | 7.1 (1.1) | 5.9 (0.9) | 6.0 (0.8) | 4.9 (0.9)* | 9.3 (1.0) | 8.05 (0.9) | 7.79 (0.8) | 7.37 (0.9) | <0.01 | 0.07 | 0.79 |
| CD-RISC 25 | 0.9 | 50.8 (3.4) | 61.6 (4.4) | 63.9 (4.0) | 66.4 (3.8)* | 51.2 (3.3) | 56.2 (4.3) | 57.3 (3.9) | 57.1 (3.7) | <0.01 | 0.27 | 0.20 |
| PSS | 0.9 | 20.3 (1.7) | 15.5 (1.7) | 15.5 (1.6) | 15.4 (1.5)* | 19.0 (1.6) | 18.6 (1.6) | 15.5 (1.5) | 14.6 (1.5)* | <0.01 | 0.91 | 0.14 |
| CompACT TOT | 0.9 | 87.5 (5.3) | 96.8 (5.4) | 100.6 (4.9) | 99.7 (5.4)* | 79.6 (5.2) | 86.2 (5.2) | 90.1 (4.8) | 89.9 (5.2)* | <0.01 | 0.15 | 0.94 |
| CompACT-OE | 0.9 | 28.83 (3.1) | 36.3 (2.5) | 38.9 (2.7) | 37.8 (2.8)* | 27.9 (3.0) | 31.4 (2.5) | 34.2 (2.6) | 32.3 (2.7) | <0.01 | 0.26 | 0.42 |
| CompACT-BA | 0.8 | 20.5 (1.7) | 21.8 (1.7) | 22.7 (1.7) | 23.7 (1.9)* | 18.1 (1.7) | 19.4 (1.7) | 20.0 (1.7) | 21.3 (1.8) | <0.01 | 0.28 | 0.99 |
| CompACT-VA | 0.8 | 38.2 (1.7) | 38.7 (2.1) | 41.2 (2.8) | 38.2 (1.9) | 35.2 (1.7) | 35.4 (2.0) | 35.9 (2.7) | 36.4 (1.8) | 0.66 | 0.21 | 0.53 |
| MAAS | 0.9 | 63.8 (3.2) | 65.5 (3.0) | 66.2 (3.1) | 64.1 (3.5) | 59.0 (3.1) | 63.2 (2.9) | 64.9 (3.0) | 65.0 (3.4) | 0.25 | 0.62 | 0.70 |
| VLQ TOT | 0.7 | 54.7 (3.2) | 59.7 (3.9) | 61.6 (4.2) | 58.4 (3.5) | 51.2 (3.1) | 51.9 (3.7) | 56.5 (4.1) | 52.3 (3.4) | 0.02 | 0.21 | 0.68 |
| VLQ Importance | 0.7 | 76.8 (2.9) | 77.1 (3.0) | 76.6 (3.1) | 76.7 (2.7) | 73.4 (2.8) | 73.8 (3.0) | 74.7 (3.0) | 73.3 (2.6) | 0.95 | 0.41 | 0.90 |
| VLQ Consistency | 0.7 | 69.6 (3.1) | 73.2 (3.3) | 75.5 (3.6) | 74.2 (3.5) | 66.2 (3.0) | 66.0 (3.3) | 75.4 (3.5) | 69.8 (3.4) | <0.01 | 0.30 | 0.51 |
| AAQ-II | 0.7 | 32.9 (2.2) | 31.5 (2.4) | 29.4 (2.2) | 27.6 (1.9) * | 33.6 (2.1) | 34.6 (2.3) | 31.6 (2.1) | 32.8 (1.9) | <0.01 | 0.30 | 0.29 |
| DDS | 0.9 | 22.5 (2.1) | 27.5 (2.0) | 28.2 (2.1) | 29.9 (2.2)* | 21.7 (2.1) | 24.1 (2.0) | 26.1 (2.0) | 26.3 (2.2)* | <0.01 | 0.36 | 0.50 |
| SEIQOL-DW | N/A | 68.4 (3.6) | N/A | 75.5 (3.4) | 72.4 (3.4)* | 64.0 (3.5) | N/A | 67.1 (3.4) | 71.2 (3.3) | 0.03 | 0.23 | 0.35 |

Data are reported as *mean* (*SE*) based on repeated measure analysis; α = Cronbach's alpha; T = time effect; R = randomization group effect; TxR = time per randomization group effect; MSQOL-54 MHC = Mental Health Component 54 item MS QoL; MSQOL-54 PHC = Physical Health Component 54 item MS QoL; HADS-D = Hospital Anxiety and Depression Scale–Depression; HADS-A = Hospital Anxiety and Depression Scale–Anxiety; CD-RISC 25 = Connor-Davidson Resilience scale; PSS = Perceived Stress Scale; CompACT TOT = Comprehensive Assessment of Acceptance and Commitment Therapy processes Total Score; OE = Openness to experience; BA = Behavioral Awareness; VA = Valued Action; MAAS = Mindful Attention Awareness Scale; VLQ TOT = Valued Living Questionnaire; AAQ-II = Acceptance and Action Questionnaire II; DDS = Drexel Defusion Scale; SEIQOL-DW = Evaluation of the Individual Quality of Life-Direct Weighting; N/A = Not applicable.

* Within-group time effect at p < 0.01

§ Within-group time effect at p < 0.05.

at each time-point. However, the standard error was high and no statistically significant differences were detected. Interestingly, despite there being no statistically significant difference was detected between READY and relaxation, the CompACT Valued Action subscale (Fig 2E) showed a trend difference between the two arms in favor of READY at T2, but it was not maintained at T3. With regards to the AAQ-II scores (Fig 2F) READY showed a progressive decrease (improvement), while relaxation is characterized by fluctuations with a consistent increase (worsening) after T2 (program completion).

The minimal clinically significant change (≥5) on the MSQOL-54 MHC was reached by 24/37 participants (11 relaxation, 57.9%; 13 READY, 72.2%), with no statistically significant differences between the two interventions ($\chi^2 = 0.8$; p = 0.36).

Participant feedback on READY was very positive. The reported level of homework engagement was good (mean = 3.5; SD = 1.0). Only 15.7% of participants reported low levels of homework completion (scored as 1 or 2). All mean program satisfaction ratings were close to the highest rating of five: global satisfaction (mean = 4.8; SD = 0.5), helpfulness (mean = 4.9; SD = 0.3), enjoyment (mean = 4.9; SD = 0.3), and usefulness of the READY Personal Plan (mean = 4.8; SD = 0.5). Participants rated very highly the extent to which the program had increased their resilience (mean = 4.4; SD = 0.7) with 72.2% of the sample reporting that they learned to better manage MS. All the participants reported that READY had positive impacts on how they feel, think about, or manage their life.

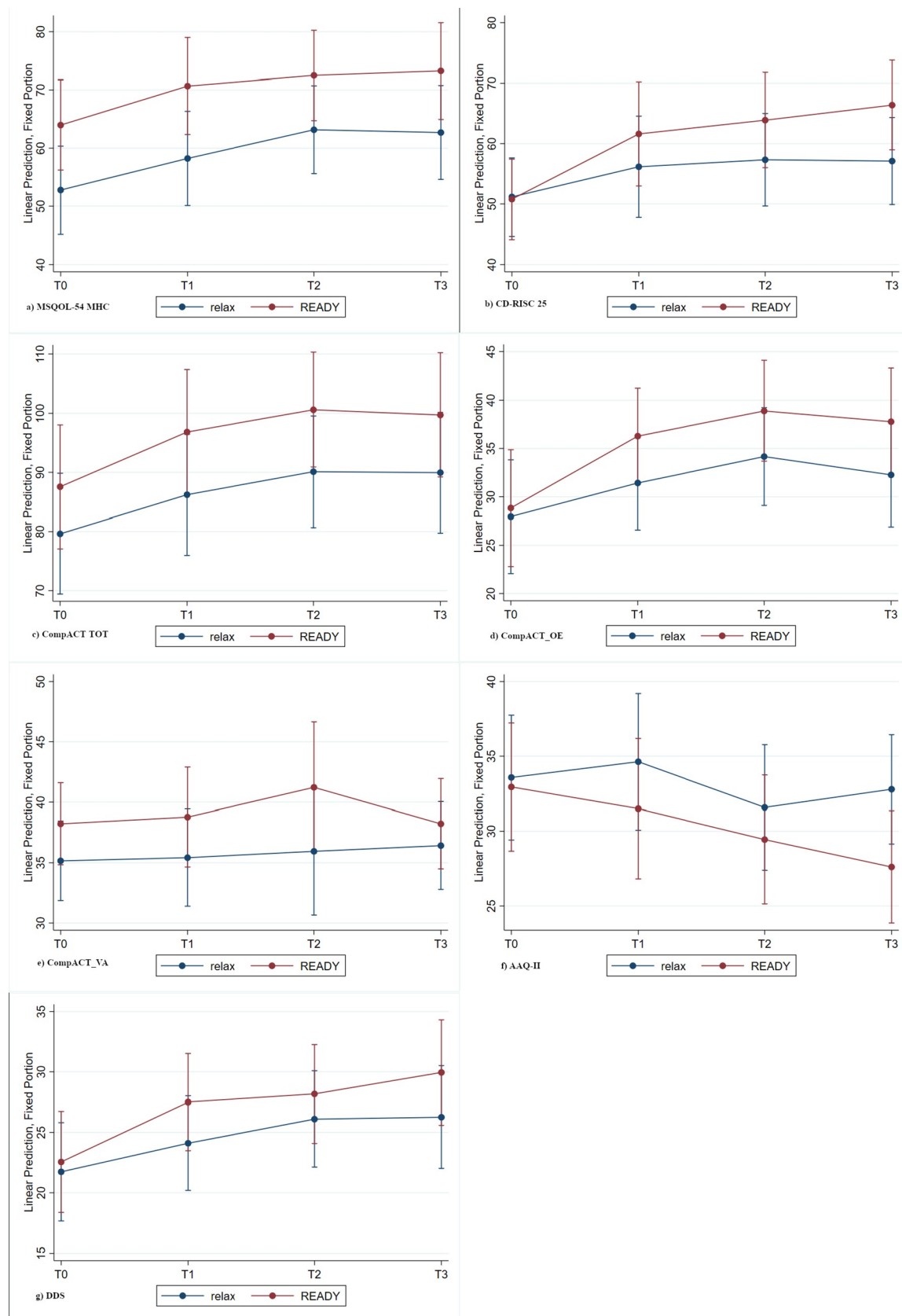

**Fig 2. Repeated measure analysis of MSQOL-54 MHC, CDRISC CompACT TOT, CompACT-OE, CompACT- VA, AAQ-II, DDS.**

### Nested qualitative study results

Fourteen relaxation participants elected to participate in the READY program after completion of the follow-up assessment. Four refused for work commitments, and one because of pregnancy. Of the total 32 participants who participated in a READY group, 30 accepted to be interviewed (18 READY, 12 relaxation + READY; mean duration of interviews = 33.5 minutes, range 16–47; 25 in person). Two refused because they reported feeling overwhelmed by the RCT assessments. The demographic and clinical characteristics of the sample are individually presented in S7 Appendix.

Content analysis of interview data revealed seven overarching themes: "Attitudes towards participation"; "Perceptions of program composition"; "Program impacts on life domains"; "Program active elements"; "Program improvement trajectories"; "Program differences and similarities"; "Suggested READY improvements". Results are presented by theme, with only the most relevant quotes to illustrate their derivation. Each quote is followed by a code in brackets reporting a participant's identification number, *gender*, *age*, *MS type*, *EDSS score*, *group allocation (R = READY, r+R = relaxation + READY)]*. The complete list of quotes for each sub-category is reported in S7 Appendix.

**Theme 1—Attitudes towards participation.** All participants reported a positive attitude towards the project together with some worries about this new experience. They believed their participation in the project was an opportunity to overcome their limits, meet other PwMS, learn new skills and increase self-care (Table 3; S7 Appendix, Table a. Attitudes towards participation):*"It was a bet with myself; I've never participated in a group therapy before!" [P14*: man, *34 y*, *RR*, *EDSS 1.0*, *R]*.

**Theme 2—Perceptions of program composition.** This theme provides insight into participants' satisfaction with the interventions (whether READY or relaxation), their formats and materials (Table 4).

READY participants were unanimously satisfied with the program: *"This experience has been enlightening! I was totally stuck, I got totally beached on my past, thoughts and worries. I had no idea how to deal with my life. The READY program helped me in starting my life again!"* [P18: woman, *41 y*, *RR*, *EDSS 2.0*, *R]*. They suggested that the READY program be included in hospital services and said they would recommend READY to ill and non-ill people: "*The*

**Table 3. Attitudes towards participation.**

| ATTITUDES TOWARDS PARTICIPATION |
| --- |
| • Curiosity |
| • Scepticism |
| • Suspiciousness |
| • Looking for a group setting experience |
| • Meeting other people with multiple sclerosis |
| • Fear |
| • Desire |
| • Put yourself out there (openness to experience) |
| • It was a bet with myself |
| • Dedicating time to yourself |
| • Dealing better with multiple sclerosis |

**Table 4. Perceptions of program composition.**

| THE PROGRAM | | THE FORMAT | | THE MATERIALS | |
|---|---|---|---|---|---|
| **READY** | **Relaxation** | **READY** | **Relaxation** | **READY** | **Relaxation** |
| *Positive* | *Positive* | • OK as is | • OK as is | • Useful | • Useful |
| • Really helpful | • Really helpful | • Prefer longer program | • Prefer shorter program | • Commitment style | |
| • Different from other therapies | • Pleasant | | | • High commitment during the program | |
| • Cutting-edge | *Negative* | | | • High commitment during and after the program | |
| • Time to dedicate to myself | • Not relaxing | | | • High commitment during program and then decrease | |
| • Helps me to be open | • A hard experience | | | • Low commitment during program but use materials after its conclusion | |
| • Good example of integrating medical and psychological care | • Passive | | | • No commitment | |
| • Should be integrated into hospital services | • Only an ice breaking activity | | | • Importance of the commitment style | |
| • Recommended for ill and non-ill persons | | | | • During the program | |
| | | | | • After the program | |
| | | | | • During and after the program | |

*medical intervention should always be integrating with a psychological one! You really need that and this is what I have found thanks to this experience!" [P16: woman, 45 y, RR, EDSS 2.0, R].*

Relaxation participants, instead, reported both positive (i.e. helpful ad pleasant) and negative opinions (i.e. not really relaxing, a hard experience, passive, and nothing more than an ice breaking activity): *"I am sorry to say that but it [relaxation] has been a hard experience! Time was running slow!" [P19: man, 57 y, RR, EDSS 2.0, r+R].* The complete list of quotes is reported in S7 Appendix, Table b. The program.

In both READY and relaxation some participants were completely satisfied with the format (S7 Appendix, Table c. The format). However, the majority of the READY participants suggested the duration of the program should be increased with a few also recommending that the duration of each session be increased: *"I wish it had lasted longer, we [group members] discussed about that and established that it would be great to increase the number of sessions from 7 to 10. You need some more sessions for ice breaking." [P07: woman, 20 y, RR, EDSS 0.0, R].* It was the opposite for relaxation, with some participants stating that the duration of the program should be reduced. *"I wish it [relaxation session] could finish fast" [P19: man, 57 y, RR, EDSS 2.0, r+R].*

Both READY and relaxation materials were judged as useful and adequate. Only READY participants spoke about their commitment to practicing new skills and using the materials at home. Although different commitment styles were reported (from high and constant to very low or absent), the majority believed commitment was crucial to benefitting from the program. There was no agreement on when commitment is more important: during the program, once the program is completed, and both during and after the program (S7 Appendix, Table d. The materials).

*"Good will is what counts! It is really important to dedicate space to practice what you learn during the sessions. Life can be frenetic, but sometimes we use that as an excuse!" [P03: woman, 39 y, RR, EDSS 2.0, R]*

Table 5. Program impacts on life domains.

| IMPACTS ON LIFE DOMAINS | |
|---|---|
| READY | Relaxation |
| • Daily living | • Multiple sclerosis (only magnetic resonance imaging management) |
| • Multiple sclerosis | • Working activities |
| • Working/academic activities | • Family |
| • Relationships | • Self-care |
| • Self-care | • Sleeping |
| • Self esteem | • Level of tension |
| • Personal growth | • Anxiety |

**Theme 3 –Program impacts on life domains.** Both READY and relaxation positively impacted the same four life domains (i.e. MS, work, relationships and self-care). However, relaxation impacts were limited to symptom management (i.e. magnetic resonance imaging management, sleeping, tension and anxiety), while READY had broader effects (i.e. daily living, self-esteem and personal growth), see Table 5. In fact, the READY experience was described as a positive "jolt", an opportunity to start savoring life again, and prompting change in how a person relates to both inner and external events: *"This experience* [READY] *has given me a new key joke for my life! When I had an argument with someone it was a real drama, an existential issue, a cosmic event! Now I am able to calm down and look at it differently, because I understand it is something that is happening in a specific context and in a precise moment. This perspective makes all the difference!" [P12: woman, 53 y, RR, EDSS 3.0, R]* (S7 Appendix, Table e. Program impacts on life domains).

**Theme 4 –Program active elements.** Specific and nonspecific therapeutic factors contribute in explaining the effects of both READY and relaxation programs (Table 6).

The specific factors reported by participants coherently reflected the approaches used (autogenic training, or resilience training based on ACT). However, relaxation participants identified fewer specific factors. In fact, only one participant was able to clearly state that her improvements were due to the autogenic training technique she learnt: *"I put in place relaxation techniques at work. I found my work environment really hard, I use autogenic training daily and I reached good results!" [P27: woman, 57 y, RR, EDSS 3.0, r+R]*, and this was also the only factor reported by relaxation participants. More nonspecific than specific factors were reported by relaxation participants. In contrast, the majority of READY participants were able to identify specific factors responsible for the impacts of the program. In fact, they clearly stated that they acquired new skills that increased their resilience: *"Thanks to the READY program I learnt to be more resilient, I am more in contact with the here and now, I give myself time and I listen to my sensations more carefully, I observe myself more; I am more in contact with my personal values and do my best to pursue them with my behaviour." [P01: woman, 53 y, RR, EDSS 2.0, R].* These new skills can be described in terms of improvement in psychological flexibility and its protective factors (present moment awareness, openness to experience, defusion, observing self, connection with personal values and values driven committed actions): *"The READY program has had a huge impact on me. I needed a shock to improve my life. Thanks to the READY program, I have learnt a new way to live my life. I am more committed in everything I do. I am actively cultivating my social relationships while I totally gave up before READY. I resumed riding the bicycle and going for a walk. I am learning how to take care of myself and I felt deeply satisfied by all these small things. I felt better at both professional and family levels because I am present in whatever I do. Dedicating time to cultivating friendship made me feel satisfied. Finding*

**Table 6. Program active elements.**

| SPECIFIC FACTORS | | NONSPECIFIC FACTORS | |
| --- | --- | --- | --- |
| **READY** | **Relaxation** | **READY** | **Relaxation** |
| • Psychological flexibility | • Autogenic training | • Group effect | • Group effect |
| • Present moment awareness (Mindfulness) | | • To see eye to eye | • To see eye to eye |
| • Formal practice | | • Sharing opinions and points of view | • Sharing opinions |
| • Informal practice | | • Feeling of "we-ness" | • Meeting people with more severe multiple sclerosis |
| • Openness to experience (Acceptance) | | • Social support | • Facilitator |
| • Defusion | | • Group as reference point | • Facilitation style |
| • Observing self | | • Feeling cared for | • Self-care |
| • Connection with personal values | | • Meeting other people with multiple sclerosis | |
| • Values driven committed actions | | • Meeting people with more severe multiple sclerosis | |
| • Social connectedness | | • Facilitator | |
| • Self-care | | • Good connection with the facilitators | |
| • Leisure activities | | • Facilitation style | |
| | | • "Practice what you preach" | |
| | | • Personal Factors | |
| | | • Willingness | |
| | | • Commitment | |

*time to chat with them without being overwhelmed by thoughts, or making a call I intended to do, are small actions that helped me in finding meaning to my days." [P18: woman, 41 y, RR, EDSS 2.0, R].* READY participants reported being more in contact with the present due to both formal and informal practice of intervention strategies. They valued acceptance because it enabled them to be open to their inner experiencing whether pleasant or not. Also identified as important resources were observing thoughts rather than taking them literally, and being in contact with a sense of self that is continuous and provides flexible perspective taking. Being in contact with personal values and acting coherently with them was highlighted by many READY participants. Parallel to psychological flexibility, people reported improvements in three other key aspects of resilience: social connectedness, self-care and leisure time activities (S7 Appendix, Table f. Specific factors).

Both READY and relaxation participants identified similar nonspecific factors including: being part of a group, sharing the same health condition, and the facilitator's style (S7 Appendix, Table g. Nonspecific factors).

> *"Being part of a group, sharing experience with others and focus on these interactions and myself for a few hours was something new. This is what I really found positive, not the relaxation technique per se." [P23: woman, 40 y, RR, EDSS 2.0, r+R]*

> *"I found the READY program really useful, first of all for the opportunity to meet other people and share our experiences, second for all the strategies we learnt that you can use in your everyday life and for dealing better with the disease." [P07: woman, 20 y, RR, EDSS 0.0, R]*

Willingness and commitment were also reported as personal factors required to be able to benefit from the READY program: *"I think you need to be open to this experience. You can't obtain the desired outcome, if you are not willing to be open to what will happen" [P07: woman,*

*20 y*, *RR*, *EDSS 0.0*, *R]*. Interestingly, one relaxation participant stated that she did not derive any benefits from relaxation directly, but by participating in the program she became committed to self-care:*"Relaxation is not my thing*! *During the relaxation training I understood I would have never practiced at home. Hence*, *I decided to dedicate time to myself. Instead of practicing relaxation*, *I focused on self-care*! *I started doing sport*, *going out with friends*, *etc . . . These activities were so much more satisfying than relaxation could be*! *Taking care of myself positively affected me."* [P23: *woman*, *40 y*, *RR*, *EDSS 2.0*, *r+R]*

**Theme 5—Program improvement trajectories.** All participants reported perceived improvements due to their participation in the study. Those who participated in both relaxation and READY reported difficulties in distinguishing between the effects of the two programs. Common improvement trajectories were detected. Some participants reported progressive improvement that continued after program completion, while others reported that the improvement effect was brief (Table 7; S7 Appendix, Table h. Program improvements trajectories.)

Interestingly, some relaxation plus READY participants reported progressive improvement during and after the programs and highlighted particular insights gained during READY: *"A lot of changes happened while I was doing this experience, particularly during the READY program. Other aspects appeared or were consolidated after it was concluded. It's a work in progress, I really want to cultivate this richness more and more."* [P27: *woman*, *57 y*, *RR*, *EDSS 3.0*, *r+R]*. One participant stated that READY may be more effective if completed after doing the relaxation program, and other participants clarified this point by suggesting that relaxation facilitated the creation of the group and READY capitalized on this and delivered resilience skills training: *"The first program [relaxation] let us to create a connection between us and the feeling of being part of a group. It really facilitates the READY. Relaxation allows you to let your tension go, the READY program instead, it is more about the techniques. The first part is about building up the group, the second one really teaches you new concepts and new skills."* [P29: *man*, *51 y*, *SP*, *EDSS 7.0*, *r+R]*

**Theme 6—Program differences and similarities.** Participants who received both relaxation and READY (N = 12) were able to identify program differences and similarities by direct comparison. Three subthemes were identified: perceived superiority of one program over the other, perceived similarities, and difficulties in distinguishing between the two (Table 8).

The majority of this sub-sample reported marked differences between READY and relaxation highlighting the superiority of READY over relaxation (S7 Appendix, Table i. Superiority of one program over the other):*"There is a substantial difference between READY and relaxation. Relaxation is like receiving a massage, while READY is like a training where you systematically learn how to cook. Everybody has some cooking abilities, but usually you have no rules or directions. During READY you acquire skills. They [READY and relaxation] are totally different!"* [P30: *woman*, *49 y*, *RR*, *EDSS 2.0*, *r+R]*. This superiority concerned three dimensions:

**Table 7. Program improvements trajectories.**

| READY | Relaxation + READY |
|---|---|
| • Illumination and further strengthening | • Improvement and further decrease |
| • Improvement and further decrease | • Progressive improvement since deciding to participate in the project |
| • Progressive improvement during and after the training | • Progressive improvement during and after the programs with a particular insights during READY |
| • The improvement can be recognized after the training | • Relaxation facilitated the creation of group, which READY capitalized on with skill development |
| | • READY is more effective if you do the relaxation before it |

**Table 8. Program differences and similarities.**

| SUPERIORITY OF ONE PROGRAM OVER THE OTHER | | PROGRAM SIMILARITIES | | LACK OF DISTINCTION BETWEEN READY AND RELAXATION |
|---|---|---|---|---|
| **READY superiority** | **Relaxation superiority** | **Shared strengths** | **Shared weaknesses** | |
| • Higher satisfaction | • Skills acquisition is easier | • Pleasant | • Program format | • Difficulties in distinguishing between the two programs |
| • More pleasant | | • Useful | • Session interval | • Programs on a continuum |
| • More engaging | | • Appropriate format | • Prolong the programs by increasing session | • Confusion between relaxation and mindfulness |
| • More interesting | | • Appropriate materials | • Interval | • Impact described as general effect of participating in the project |
| • More useful | | • Acquisition of new strategies | • Number | • Daily living |
| • More detailed and deeper | | • I would recommend them | • Suboptimal session time | • Multiple sclerosis |
| • More involving | | • Group effect | | • Magnetic Resonance Imaging |
| • More self-disclosure | | • Meeting other people with multiple sclerosis | | • Work activities |
| • Richer topics | | | | • Relationship |
| • More active | | | | • Patient-neurologist relationship |
| • Better setting | | | | • Personality |
| • I keep practicing only READY strategies | | | | • Openness to the experience |
| • I would suggest only READY | | | | • Reducing panic attacks |
| • Impact on more life areas | | | | |
| • Daily living | | | | |
| • MS | | | | |
| • Working activities | | | | |
| • Family | | | | |
| • Self-efficacy | | | | |
| • Self esteem | | | | |
| • Personal balance | | | | |
| • Acquisition of more skills | | | | |
| • Present moment awareness | | | | |
| • Openness to experience | | | | |
| • Defusion | | | | |
| • Connection with personal values | | | | |
| • Values driven committed actions | | | | |

higher satisfaction with READY; READY positively impacted more areas than relaxation; READY enabled the acquisition of more skills than relaxation. Considering this last category, one participant stated: "*Differently from relaxation, during the READY program I acquired new skills such as: mindfulness, defusion and acceptance. Learning new strategies has helped me in felling the punch less strong. This is really important.*" *[P19: man, 57 y, RR, EDSS 2.0, r+R]*. People described this improvement in terms of increased resilience: "*I notice the results of the READY every day. I am more resilient!*" *[P22: man, 49 y, RR, EDSS 4.0, r+R]*, and they identified the psychological flexibility protective factors (present moment awareness, openness to experience, defusion, connection with personal values, and committed action) as mediators of this

effect. The only advantage of relaxation over READY was that the relaxation technique was perceived as easier to acquire than the READY skills: "*Learning READY skills is harder than relaxation. It was easier with relaxation technique.*" [P24: *woman, 35 y, RR, EDSS 0.0, r+R*]

Despite the two programs not being perceived as equivalent, some participants reported similarities between the two in terms of shared strengths or weaknesses (S7 Appendix, Table l. Program similarities). Common strengths included: pleasant, useful, adequate format and materials, delivery of new skills and group processes. Some participants enjoyed the experience so much that they wished it could last longer: "*It was such a beautiful experience that we were so sad when it finished! For this reason I would suggest to extend the number of the session... maybe it is not really necessary, but we all would like the program could last longer.*" [P20: *man, 55 y, RR, EDSS 2.0, r+R*]. Proposed solutions varied from increasing the number of sessions to increasing the interval between sessions.

Some participants were not able to clearly distinguish between the two programs with several describing them as a continuum (S7 Appendix, Table m. Lack of distinction between READY and relaxation): "*I can't split up the two experiences. We passed from one experience to the other one. I embraced this experience all together, as a whole.*" [P27: *woman, 57 y, RR, EDSS 3.0, r+R*]. Others perceived mindfulness and relaxation practice as being the same: "*The relaxation/mindfulness program... I don't divide them. I think mindfulness and relaxation are really close to each other. To me relaxation means practicing mindfulness.*" [P25: *man, 57 y, RR, EDSS 2.0, r+R*]

**Theme 7 –Suggested READY improvements.** Themes and categories that emerged from responses to questions on how the READY program can be improved are presented in Table 9 with the number of interviewees who reported it, in order to track the amplitude of each recommended change.

All participants were very satisfied with the program: *Everything was ok, the number of session, the duration of the program, everything! We were scared about cutting the cord, but in the end, you need to start walking by your own. We learnt so many skills that we need practicing them even without supervision [P01: woman, 53 y, RR, EDSS 2.0, R]*. However, more than half of the sample also suggested some improvements, that fell into six categories: 1) Program format and materials; 2) Setting; 3) Topics; 4) Procedures enrolling patients in the programs; 5) Planning dedicated meetings; 6) Offering a READY for MS group to patient's significant others (S7 Appendix, Table n. Suggested READY improvements).

## Discussion

This pilot RCT evaluated the feasibility and acceptability of the Italian READY for MS program and its efficacy by comparing it with an active control condition (relaxation). Consistent with the literature on psychosocial interventions in PwMS [62], most participants reported significant improvements in several psychological dimensions at the three month follow-up, with 65% reaching a clinically significant improvement in the mental component of health-related QoL (primary outcome). In addition, the improvements evident in the READY sample are in line with those shown in prior resilience training research with MS and other medical conditions [63,64], as well as with previous studies that have evaluated the READY program [27–30].

Both qualitative and quantitative data showed that all READY participants were satisfied with the program and its materials, and the majority had a good level of engagement. All READY participants stated that it increased their resilience and positively affected their life, and the majority declared it helped them in dealing better with MS. Noteworthy is the wide variation in disease severity (EDSS) among the READY participants which supports its utility

**Table 9. Suggested READY improvements.**

| SUGGESTED READY IMPROVEMENTS |
| --- |
| • Program format and materials |
| • Add booster/Recall online (11) |
| • Increase number of sessions (14) |
| • Increase duration of sessions (2) |
| • Increase time between sessions (6) |
| • Digitalize the materials (1) |
| • A dedicated repository with real life models of READY skills application (1) |
| • Setting |
| • Physical Setting |
| • Outside the Multiple Sclerosis Centre (1) |
| • Quieter room (3) |
| • Group composition |
| • Age (1) |
| • Disease severity (1) |
| • Gender (1) |
| • Bigger group (2) |
| • Individual sessions with facilitator (2) |
| • Topics |
| • Add new topics (2) |
| • Expand on current topics (3) |
| • Procedure for enrolling patients |
| • Pre-program interview with facilitator (2) |
| • Offering READY at the beginning of the patients' health care pathway (1) |
| • Additional events |
| • For members across groups (4) |
| • For presenting study results (2) |
| • For family of people with multiple sclerosis (3) |
| • Offering READY to patients' significant others (7) |

The number of interviewees reporting each category out of the total sample of 30 is noted in brackets

and acceptability in this population. All participants said that they would recommend the READY program to others with MS, and some suggested it should be offered to people with other medical conditions and carers.

Despite the high level of satisfaction with the READY program, it was not more efficacious than relaxation in evidencing statistically significant improvements on the outcome measures. This is in line with a previous study on MS outpatients with anxiety and/or depression, where between-group comparisons did not demonstrate any advantages of ACT over relaxation on any outcome variables [65]. That pilot study did not collect qualitative data on participants' experience and views of the interventions, despite literature highlighting the importance of using both qualitative and quantitative methods in RCTs evaluating complex interventions [66, 67].

One interpretation of the lack of superiority of READY over relaxation in the statistical analyses, is that the READY and relaxation interventions are equally effective in improving the psychological functioning of PwMS. This is consistent with the "Dodo Bird Verdict", which suggests that the equivalence may be due to common factors among different psychotherapies [68]. Participants who received both READY and relaxation interventions identified the

following two common factors: the group setting and the facilitation style. In addition, although mindfulness (a key intervention mechanism of the READY program) and relaxation are different constructs and in practice different techniques, there is nevertheless some overlap. The similarities between the two are supported by the qualitative data, with one of the themes indicating that it was difficult for some participants to distinguish between the two interventions and with some participants specifically stating that they viewed the relaxation and mindfulness techniques as being similar. It is likely that having one ACT trained facilitator for both interventions further exacerbated the overlap between these two techniques and the commonalities between the two programs. Specifically, it is possible that the delivery of the relaxation intervention may have been contaminated with a bias towards ACT even though separate manuals were used to guide facilitation. No intervention fidelity data were collected in real time to test this proposal. These issues will be addressed in the multi-centre RCT by having different facilitators for the READY and control group interventions and by audio recording each session.

The proposal that READY and relaxation are equally effective is undermined by the results of the nested qualitative study in which participants reported the superiority of READY over relaxation in terms of satisfaction, impact on life domains, and the range of skills acquired. Drawing on empirical evidence from a phase I and II anti-cancer drug trial, Cox demonstrated how different methods of collecting data on QoL (patient-reported outcome measures and interviews) can lead to divergent conclusions about participants' trial experiences and intervention impacts on their QoL [69]. As reported by Moffat et al, it is not surprising that quantitative and qualitative methods may yield divergent findings since each explores different, but related questions, and both are based on different theoretical paradigms [70]. Despite in the MRC framework qualitative analysis serves to better understand the quantitative findings of the RCT, it is interesting to note that Brannen [66] and Bryman [67] caution against simply combining results from the two methods. Moffat et al. suggested managing qualitative and quantitative data as complementary rather than in competition for identifying the true version of events [70]. Therefore, the quantitative findings from our pilot RCT should be carefully considered in the context of the qualitative data.

In line with this, the absence of statistically significant intervention effects of READY may be accounted for by three methodological limitations: the small sample size, the short follow-up, and a ceiling effect with the primary outcome. Regarding sample size, this was a pilot study, with only the power to detect large differences, if present. Resilience, acceptance (as measured by both AAQ-II and the CompACT openness to experience subscale) and defusion score plots showed that READY participants improved more than those in the relaxation condition. However, the standard error was high, and no statistically significant differences were detected. A bigger sample size would decrease the standard error and consequently increase the study reliability. With respect to the short follow-up, READY participants not only improved more than relaxation participants on these scales, the differences between the two arms also increased at each time-point, showing a promising longitudinal trend (particularly for resilience). It is possible that the three month follow-up was not long enough to capture the accumulative process of change activated by the READY program and that with a longer follow-up this trend would become significant. For these reasons, the multi-centre RCT will be designed with a longer follow-up. Finally, a ceiling effect may have occurred with regards to the primary outcome measure [71]. In fact, more than one fourth of all participants had elevated mental health QoL scores with the mean of the READY condition being significantly higher than that of the relaxation condition at baseline. A bigger sample size will assure a more balanced allocation of participants in terms of primary outcome scores at baseline.

At a qualitative level, participants reported that READY was more intense and comprehensive than relaxation. Interestingly, qualitative data and satisfaction ratings provided support

for the hypothesized mediators of change in READY–the six ACT processes. In fact, while relaxation participants strongly suggested the beneficial effect was mainly due to nonspecific factors such as being in a group, the facilitator style and taking care of themselves, READY participants accurately described both nonspecific and specific factors and recalled each of the ACT processes (acceptance, cognitive defusion, mindfulness, self as context, connection with personal values, values driven committed action).

Although participants were highly satisfied with READY and its format, they suggested several changes including increasing program duration via: adding more booster sessions ideally in person or at least online; increasing the number of sessions or their duration; increasing the time between sessions; digitalizing the materials to augment the group sessions. This result differs from that of the Australian study where the original format was confirmed by participant feedback [30]. The differing results in this regard may be due to cultural differences.

In addition to the methodological weaknesses mentioned above, the following study limitations should also be considered. First, participants were enrolled from the same MS Centre. Second, with the exception of the SEIQOL-DW, the outcomes consisted of patient-reported measures. The study may benefit from a clinical evaluation by an independent assessor. Third, control group participants were offered the READY program immediately after the last follow-up evaluation, which may have had a positive effect on their mood which in turn positively biased their self-report on the READY program. Fourth, although the facilitator (AMG) had extensive training in both ACT and READY, received supervision by one of the program developers (KP) and used the manual to deliver the program, intervention fidelity was not assessed via audio/video recording. Fifth, despite the wide EDSS score range (0–6.5), the median EDSS score was 2.0 denoting that the majority of PwMS had low neurological impairment. However, it is also important to note that the relationship between physical function and resilience in MS is still debated in literature [72–75] and seems not to play a major role in independently predicting resilience [76]. Further studies should also explore if MS type (RR or progressive) may influence intervention effects. In addition, we did not control for premorbid characteristics (for e.g., cognitive reserve, intelligence level and core personality characteristics) of participants.

## Conclusion

The READY program was well accepted by PwMS with varied socio-demographic and clinical characteristics, suggesting it has high utility and acceptability in an Italian clinical setting. Statistical analyses showed that READY was not more efficacious than relaxation. In contrast, qualitative data indicated that READY was viewed by participants as superior to relaxation; a finding that converged with four non-significant statistical trends supporting the efficacy of READY. Consistent with the ACT psychological flexibility framework, the qualitative data indicated that participants' perceived improvements in resilience and health-related QoL were due to the acquisition of skills related to the six core ACT processes.

The Steering Committee and an international expert panel (four persons) jointly discussed the study findings in two dedicated meetings (13 and 24 January 2020) and outlined the multi-centre RCT evaluation of the Italian READY for MS program. It was a structured discussion, using a "PICO" (Population, Intervention, Comparator, Outcomes) format [77]. The eligibility criteria of the pilot RCT, as well as the READY program were deemed adequate by the panel (Population and Intervention). To limit possible involuntary contaminations, the multi-centre RCT control group will be conducted by a psychologist not involved in "READY for MS training program" and with no expertise in ACT or mindfulness interventions (Comparator). Following Chmitorz et al. [78] and based on the trend differences in the CD-RISC 25 [52]

observed in the pilot RCT, change in resilience at three month follow-up will be the primary trial endpoint. In addition, a longer follow-up assessment scheduled six months post-intervention was set to evaluate effect maintenance (Outcomes).

## Supporting information

**S1 Appendix. CONSORT.**
(PDF)

**S2 Appendix. COREQ.**
(PDF)

**S3 Appendix. Study protocol.**
(PDF)

**S4 Appendix. Interview guide.**
(PDF)

**S5 Appendix. Sample size calculation.**
(PDF)

**S6 Appendix. Graphic plots for repeated measure analysis.**
(PDF)

**S7 Appendix. Qualitative study quotes.**
(PDF)

## Acknowledgments

The authors thank the PwMS who participated in the project, Sara Ducale and Jessica Argenziano for their support during group facilitation and data collection, and Prof. Bagnardi for his advice on data analyses.

## Author Contributions

**Conceptualization:** Ambra Mara Giovannetti, Alessandra Solari, Kenneth Ian Pakenham.

**Data curation:** Ambra Mara Giovannetti.

**Formal analysis:** Ambra Mara Giovannetti, Irene Tramacere.

**Funding acquisition:** Ambra Mara Giovannetti, Michele Messmer Uccelli, Alessandra Solari, Kenneth Ian Pakenham.

**Investigation:** Ambra Mara Giovannetti, Rui Quintas.

**Methodology:** Ambra Mara Giovannetti, Andrea Giordano, Alessandra Solari, Kenneth Ian Pakenham.

**Project administration:** Ambra Mara Giovannetti.

**Resources:** Ambra Mara Giovannetti.

**Supervision:** Alessandra Solari, Kenneth Ian Pakenham.

**Validation:** Alessandra Solari.

**Visualization:** Ambra Mara Giovannetti.

**Writing – original draft:** Ambra Mara Giovannetti, Rui Quintas, Andrea Giordano, Paolo Confalonieri, Michele Messmer Uccelli, Alessandra Solari, Kenneth Ian Pakenham.

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
