## [Decision Letter · Decision Letter 0]

13 Feb 2020

PONE-D-19-32115

A resilience group training program for people with MS: results of a pilot single-blind randomized controlled trial and nested qualitative study

PLOS ONE

Dear Dr Giovannetti,

Thank you for submitting your manuscript to PLOS ONE. After careful consideration, we feel that it has merit but does not fully meet PLOS ONE’s publication criteria as it currently stands. Therefore, we invite you to submit a revised version of the manuscript that addresses the points raised during the review process.

Specifically the statistical methodology and data handling should be worked out. Similarly the clinical aspects in need of clarification should be addressed. Please carefully consider all the reviewers' points.

We would appreciate receiving your revised manuscript by MArch 20th. To enhance the reproducibility of your results, we recommend that if applicable you deposit your laboratory protocols in protocols.io, where a protocol can be assigned its own identifier (DOI) such that it can be cited independently in the future. For instructions see: http://journals.plos.org/plosone/s/submission-guidelines#loc-laboratory-protocols

We look forward to receiving your revised manuscript.

Kind regards,

Andrea Martinuzzi

Academic Editor

PLOS ONE

Journal Requirements:

2. Thank you for submitting your clinical trial to PLOS ONE and for providing the name of the registry and the registration number. The information in the registry entry suggests that your trial was registered after patient recruitment began. PLOS ONE strongly encourages authors to register all trials before recruiting the first participant in a study.

1) your reasons for your delay in registering this study (after enrolment of participants started);

2) confirmation that all related trials are registered by stating: “The authors confirm that all ongoing and related trials for this drug/intervention are registered”.

Please also ensure you report the date at which the ethics committee approved the study as well as the complete date range for patient recruitment and follow-up in the Methods section of your manuscript.

"As a staff member of the University of Queensland and co-Author of the READY program, Dr. Pakenham receives royalties from UniQuest for commercial (not research) licensing arrangements entered into by third parties who want to deliver the program."

Reviewers' comments:

Reviewer's Responses to Questions

**Comments to the Author**

1. Is the manuscript technically sound, and do the data support the conclusions?

Reviewer #1: Partly

Reviewer #2: Yes

Reviewer #3: Yes

2. Has the statistical analysis been performed appropriately and rigorously? 

Reviewer #1: No

Reviewer #2: Yes

Reviewer #3: Yes

3. Have the authors made all data underlying the findings in their manuscript fully available?

Reviewer #1: Yes

Reviewer #2: Yes

Reviewer #3: No

4. Is the manuscript presented in an intelligible fashion and written in standard English?

Reviewer #1: Yes

Reviewer #2: Yes

Reviewer #3: Yes

5. Review Comments to the Author

Reviewer #1: This is an interesting longitudinal study where the qualitative substudy yielded more interesting findings than the main quantitative study. I cannot help but feel the study is underpowered, and I gives some comments on the quantitative analysis that may make it more convincing.

1. Sample size is based on a "post-intervention effect size" of 64%. I don't know what this means. A 64% improvement in the baseline QOL score? It is not clear what this effect size is for an ordinal measure like a questionnaire score.

2. Repeated measures mixed effect models are used, but I cannot find the specific models examined, the assumptions of those models (including the covariance structure and a justification), or any diagnostics (residuals, Q-Q plots, etc.) that we would require of any student in a regression class to see if these models are really appropriate. I also wonder if the sample size computation would have been more appropriate by looking at sample size for repeated measures ANOVA models instead of a simple effect change (noting that I still don't know what the effect is!). Naturally, a more complicated model will require a larger sample size. I would also be interested in seeing a single longitudinal plot by subject superimposed for intervention and control

3. The conclusions should clearly state whether the assumptions made in the sample size computation were realized in the study, and if not, what the impact of power is on the final analysis.

Reviewer #2: This is an interesting paper that evaluated the feasibility and acceptability of the Italian READY for MS program, and preliminary assessed its efficacy when compared to an active control intervention(group relaxation). Although the program was well accepted by MS patients it was not more efficacious than relaxation.

The paper is well written and methodologically sound in most respects.

The measures utilized are suitable to the population and study questions.

However, certain issues require the authors attention in order to provide generalizability of the findings

1. Were premorbid characteristics of the individuals (MS and HC) considered as covariates or found to be matched at baseline assessment (for e.g., Cognitive reserve, intelligence level, core personality characteristics). Resilience capacity may be influenced by these factors and especially cognitive flexibility and executive function capacity.

2. Participant engagement, drop out rate and satisfaction might have varied if the intervention was applied and tested in RRMS vs Progressive MS (SPMS or PPMS) patients. Progressive patients psychological, physical and cognitive state may have differentially influenced the study outcome. This should be noted as a limitation or tested if possible as this was a mixed MS sample.

3. The median EDSS score of the MS sample denotes that many participants included may still have low disability staus which may differentially impact these patients resilience compared to individuals with high disability (e.g., wheelchair bound, more severe cognitive deficits, higher depression and fatigue). This should be noted as a limitation or tested if possible be dividing patients into low EDSS < 3 and over 3

4. This was a high educated sample with 16 participants having degrees and 4 PhDs. In my opoinion this variable may have significantly influenced the outcome as these patients utilize different coping strategies to illiterate or very low education individuals. Please test if educational level influenced outcome via regression analysis in the groups (even though the groups were matched at baseline as noted on this variable)

Reviewer #3: The aim is to present the results of the pilot RCT (phase 1), which evaluates the feasibility and acceptability of the Italian READY for MS program and its efficacy by comparing it to an active control intervention (group relaxation).

The introduction is well-written and clearly defines the problem to be addressed and the purpose of the paper. I found the manuscript to be clearly presented and worthy of publication, with some minor points clarified.

Methods: it would be useful to add in full details of who collected the data (e.g. medical professional, researcher), where they did this, and how long the interview was. Currently I can only see that a clinical psychologist did the baseline measures.

Could the authors clarify why they used computer-based stratified randomization (2 factors: Expanded 166 Disability Status Scale (EDSS) [42] score < 2.0 and ≥ 2.0, and CDRISC score < 50 and ≥ 50).

Sample size – In the discussion the authors suggest the small sample size may have led to null quantitative results. The methods state the sample size calculation was based on achieving a large effect size, as reported in other studies. Can the authors provide some further discussion here? If their sample size calculation was correct, then the small sample size argument may be less acceptable.

The differences between qual and quant outcomes - could the authors offer some thoughts about this?

6. PLOS authors have the option to publish the peer review history of their article (what does this mean?). If published, this will include your full peer review and any attached files.

Reviewer #1: No

Reviewer #2: No

Reviewer #3: No

---

## [Author Response · Author response to Decision Letter 0]

11 Mar 2020

Response to Reviewers

PONE-D-19-32115

A resilience group training program for people with MS: results of a pilot single-blind randomized controlled trial and nested qualitative study

PLOS ONE

To enhance the reproducibility of your results, we recommend that if applicable you deposit your laboratory protocols in protocols.io, where a protocol can be assigned its own identifier (DOI) such that it can be cited independently in the future. For instructions see: http://journals.plos.org/plosone/s/submission-guidelines#loc-laboratory-protocols

Dear Editor, thank you for your advice. We deposited the protocol in protocols.io and it has been assigned the following DOI: dx.doi.org/10.17504/protocols.io.bcqjivun

We added this information in the text in the Materials and Methods section: “The protocol (S3 Appendix; DOI: dx.doi.org/10.17504/protocols.io.bcqjivun) received ethical clearance from the ethics committees of the Fondazione IRCCS Istituto Neurologico Carlo Besta (8 February 2017, internal ref: 37; amendment approved 06 September 2017, internal ref: 43) and The University of Queensland (8 May 2018, clearance number: 2018000942/Review 08022017).”

Authors Additional Comment:

Dear editor, 

After the paper submission to PLOS ONE, we decided to discuss the results of the pilot RCT with an international panel. The discussion was rich and much debated and it significantly impacted on the design of the Multi-centre RCT that follows. For this reason we decided to change the original conclusion paragraph of the manuscript with the one reported below (see edits in red):

“The Steering Committee and an international expert panel (four persons) jointly discussed the study findings in two dedicated meetings (13 and 24 January 2020) and outlined the multi-centre RCT evaluation of the Italian READY for MS program. It was a structured discussion, using a “PICO” (Population, Intervention, Comparator, Outcomes) format [77]. The eligibility criteria of the pilot RCT, as well as the READY program were deemed adequate by the panel (Population and Intervention). To limit possible involuntary contaminations, the multi-centre RCT control group will be run by a psychologist not involved in “READY for MS training program” and with no expertise in ACT or mindfulness interventions (Comparator). Following Chmitorz et al. [78] and based on the trend differences in the CD-RISC 25 [52] observed in the pilot RCT, change in resilience at three month follow-up will be the primary trial endpoint. In addition, a longer follow-up assessment scheduled six months post-intervention was decided to evaluate effect maintenance (Outcomes).”

Journal Requirements:

We checked all the materials and modified the formatting style where needed.

2. Thank you for submitting your clinical trial to PLOS ONE and for providing the name of the registry and the registration number. The information in the registry entry suggests that your trial was registered after patient recruitment began. PLOS ONE strongly encourages authors to register all trials before recruiting the first participant in a study.

1) your reasons for your delay in registering this study (after enrolment of participants started);

2) confirmation that all related trials are registered by stating: “The authors confirm that all ongoing and related trials for this drug/intervention are registered”.

We included the statement in the text as reported above (see edits in red): “The study was registered on the ISRCTN registry (ISRCTN registration number: 38971970), 5 June 2018, before enrolment completion. The authors confirm that all ongoing and related trials for this intervention are registered.”

Please also ensure you report the date at which the ethics committee approved the study as well as the complete date range for patient recruitment and follow-up in the Methods section of your manuscript.

Ethics committee approval was reported in the main text at the beginning of the methodology session (see edits in red): “The protocol (S3 Appendix; DOI: dx.doi.org/10.17504/protocols.io.bcqjivun) received ethical clearance from the ethics committees of the Fondazione IRCCS Istituto Neurologico Carlo Besta (8 February 2017, internal ref: 37; amendment approved 06 September 2017, internal ref: 43) and The University of Queensland (8 May 2018, clearance number: 2018000942/Review 08022017). All participants gave written informed consent. The study was registered on the ISRCTN registry (ISRCTN registration number: 38971970), 5 June 2018, before enrolment completion. The authors confirm that all ongoing and related trials for this intervention are registered.”

The complete details for patients’ recruitment and follow-up was included in the Results/RCT section as reported above (see edits in red):

“Four intervention groups were conducted (two READY and two relaxation groups) between April 2017 and January 2018 (first recruitment start date 16 March 2017, end date 30 March 2017; follow-up completed in September 2017. Second recruitment start date 9 October 2017, end date 23 October 2017; follow-up completed in April 2018). The rate of recruitment was high (70.0%) and the most common reasons for refusal were time conflicts with work. Thirty-nine patients participated; 20 were assigned to READY, 19 to relaxation. Two participants (READY) withdrew before beginning the intervention due to unexpected work commitments (See Fig. 1). 

"As a staff member of the University of Queensland and co-Author of the READY program, Dr. Pakenham receives royalties from UniQuest for commercial (not research) licensing arrangements entered into by third parties who want to deliver the program."

We modified the Conflict of Interest Statement as per editor request and added information for AS and PC (see edits in red).

“AS reports grants from Fondazione Italiana Sclerosi Multipla (FISM), during the conduct of the study; personal fees from Biogen Idec, Merck Serono, Novartis, Almirall, and Excemed. As a staff member of the University of Queensland and co-Author of the READY program, KP receives royalties from UniQuest for commercial (not research) licensing arrangements entered into by third parties who want to deliver the program. PC acted as an Advisory Board member of Biogen and Novartis; received funding for traveling from Biogen, Merck, Teva, Novartis; received honoraria for speaking or writing from Biogen and Novartis. He received support for research project by Novartis and Merk and is involved as principal investigator or co-investigator in clinical trials for Teva, Novartis, Biogen and Merk. RQ received funding for conducting research and clinical projects from AIM Education (Novartis) and Fondazione Serono (Merck) in 2019. This does not alter our adherence to PLOS ONE policies on sharing data.”

Reviewers' comments:

5. Review Comments to the Author

Reviewer #1: This is an interesting longitudinal study where the qualitative substudy yielded more interesting findings than the main quantitative study. I cannot help but feel the study is underpowered, and I gives some comments on the quantitative analysis that may make it more convincing.

1. Sample size is based on a "post-intervention effect size" of 64%. I don't know what this means. A 64% improvement in the baseline QOL score? It is not clear what this effect size is for an ordinal measure like a questionnaire score.

We thank the reviewer for the comment. This is a pilot RCT and our aim was to assess the efficacy of the Italian READY for MS program in improving QoL (primary outcome; measured with the Mental Health Component of the 54-items MS Quality of Life inventory, MHC MSQOL-54), mood (HADS; PSS), individualized quality of life (SEIQOL-DW), resilience (CDRISC-25), psychological flexibility (CompACT), and its protective factors: Acceptance (AAQ-II); Cognitive defusion (DDS); Contact with the present moment (MAAS); Values and committed actions (VLQ), compared to the control group (relaxation).

The sample size was calculated on the primary outcome (MSQOL-54 MHC). In the S5 Appendix it was stated that the sample size was estimated for two samples (i.e. arms) with repeated measures. The ‘64%’ is the effect size calculated as follows: (m2-m1)/pooled SD= 56-70/22=64%. In our calculation we used the CHANGE (mean changes) method which uses each patient’s difference between the mean of the follow-up (MSQOL-54 MHC) measurements and the mean of baseline (MSQOL-54 MHC) measurements as the summary measurement. It compares the two groups by using a simple t test.

Based on the reviewer’s suggestion, we revised the sample size section as follows (see edits in red): “A sample size of 32 patients (18 per arm) achieves 80% power to detect a difference of 14 (standard deviation 22) in the MSQOL-54 MHC in a design with three repeated measurements, assuming an intra-subject correlation between observations of 0.75, an alpha error of 0.05, and 20% of the patients lost to follow-up (S5 Appendix). Because no data on the READY for MS were available when the study was designed, our estimate was based on the large effect size for QoL (d = 0.80) assessed by the Profile of Health-related Quality of Life in Chronic Disorders scale in a RCT of a group mindfulness intervention for PwMS [58], and on available data on the MSQOL-54 MHC [49, 59]”.

2. Repeated measures mixed effect models are used, but I cannot find the specific models examined, the assumptions of those models (including the covariance structure and a justification), or any diagnostics (residuals, Q-Q plots, etc.) that we would require of any student in a regression class to see if these models are really appropriate. 

We used repeated measures mixed effects models with identity covariance structure and chose the covariance structure with the lowest Akaike information criterion (AIC). The table below reports the AIC scores on the MSQOL-54 MHC (primary outcome).

Covariance structure AIC

Identity 1176.477

Independent 1176.869

Exchangeable 1178.477

Unstructured 1183.159

We modified the analysis section as follows (see changes in red):

“Longitudinal changes were analyzed using repeated measures mixed effects models with identity covariance structure.”

Model assumptions were upheld as demonstrated by the following information (P-P plot, Q-Q plot and Kernel density graph) on the MSQOL-54 MHC (primary outcome measure). See Fig. 1-3

Fig. 1 P-P plot

Fig. 2 Q-Q plot

Fig. 3 Kernel density graph

I also wonder if the sample size computation would have been more appropriate by looking at sample size for repeated measures ANOVA models instead of a simple effect change (noting that I still don't know what the effect is!). Naturally, a more complicated model will require a larger sample size.

As clarified in our response to comment 1, we estimated the sample size for two samples with repeated measures. We hope the text is clearer now.

I would also be interested in seeing a single longitudinal plot by subject superimposed for intervention and control.

Please find the single longitudinal plot by subject superimposed for intervention and control, on the primary outcome measure (MSQOL-54 MHC). See Fig. 4.

Fig. 4 Longitudinal plot by subject for intervention and control

3. The conclusions should clearly state whether the assumptions made in the sample size computation were realized in the study, and if not, what the impact of power is on the final analysis.

In accord with your comments and those of Reviewer 3, we modified the sample size section and the discussion in order to clarify how we calculated the sample size (please see our reply to your first comment) and how the information from this pilot study guides the power calculation of the next study (see edits in the conclusion section).

Reviewer #2: This is an interesting paper that evaluated the feasibility and acceptability of the Italian READY for MS program, and preliminary assessed its efficacy when compared to an active control intervention (group relaxation). Although the program was well accepted by MS patients it was not more efficacious than relaxation.

The paper is well written and methodologically sound in most respects.

The measures utilized are suitable to the population and study questions.

However, certain issues require the authors attention in order to provide generalizability of the findings

1. Were premorbid characteristics of the individuals (MS and HC) considered as covariates or found to be matched at baseline assessment (for e.g., Cognitive reserve, intelligence level, core personality characteristics). Resilience capacity may be influenced by these factors and especially cognitive flexibility and executive function capacity.

It would be interesting to consider if the premorbid characteristics suggested (for e.g., Cognitive reserve, intelligence level, core personality characteristics) influence resilience. However, we did not assessed them. Randomization included the following two factors: Expanded Disability Status Scale (EDSS) and CD-RISC 25 score.

We added a dedicated statement in the limitation (see edits in red):

“In addition, we did not control for premorbid characteristics (for e.g., cognitive reserve, intelligence level and core personality characteristics) of participants.”

2. Participant engagement, drop-out rate and satisfaction might have varied if the intervention was applied and tested in RRMS vs Progressive MS (SPMS or PPMS) patients. Progressive patients psychological, physical and cognitive state may have differentially influenced the study outcome. This should be noted as a limitation or tested if possible as this was a mixed MS sample.

Because the sample size is limited and the number of people with relapsing remitting or progressive MS are unbalanced in our sample (81% and 19% respectively) we cannot conduct separate analyses for these subgroups with adequate reliability. We appreciate the reasoning underlying the suggestion to include it as a limitation (edits in red). 

“Further studies should also explore if MS types (RR or progressive form) may influence the program outcome.”

3. The median EDSS score of the MS sample denotes that many participants included may still have low disability status which may differentially impact these patients resilience compared to individuals with high disability (e.g., wheelchair bound, more severe cognitive deficits, higher depression and fatigue). This should be noted as a limitation or tested if possible be dividing patients into low EDSS < 3 and over 3

We added this in the limitation sessions: “Fifth, despite the wide EDSS score range (0-6.5), the median EDSS score was 2.0 denoting that the majority of the patients had low neurological impairment. However, it is also important to notice that the relationship between physical function and resilience in MS is still debated in the literature [67-70] and seems not to play a major role in independently predicting resilience [71].”

4. This was a high educated sample with 16 participants having degrees and 4 PhDs. In my opinion this variable may have significantly influenced the outcome as these patients utilize different coping strategies to illiterate or very low education individuals. Please test if educational level influenced outcome via regression analysis in the groups (even though the groups were matched at baseline as noted on this variable)

As per your request, we verified the impact of education on the longitudinal change of the primary outcome at T3. Education did not have any significant effects on changes in the primary outcome (p=0.81). Therefore we did not included the effects of education as a limitation.

Reviewer #3: The aim is to present the results of the pilot RCT (phase 1), which evaluates the feasibility and acceptability of the Italian READY for MS program and its efficacy by comparing it to an active control intervention (group relaxation).

The introduction is well-written and clearly defines the problem to be addressed and the purpose of the paper. I found the manuscript to be clearly presented and worthy of publication, with some minor points clarified.

Methods: it would be useful to add in full details of who collected the data (e.g. medical professional, researcher), where they did this, and how long the interview was. Currently I can only see that a clinical psychologist did the baseline measures.

Thanks for your suggestions. Some of the information required is included in the checklists (S1 and S2). We added more details on data collection in the text (see edits in red). In particular:

1) In Materials and Methods: “In addition, the examining clinician (not involved in intervention delivery and blinded to patient assignment) administered the Schedule for the Evaluation of the Individual Quality of Life-Direct Weighting (SEIQoL-DW) [40] at T0, T2 and T3. The SEIQoL-DW was administered in a dedicated room of the hospital. The examining clinician (RQ) was a researcher and clinical psychologist working at the MS Centre of the Fondazione IRCCS Istituto Neurologico Carlo Besta, with expertise in qualitative methodologies and specifically trained to administer the SEIQoL-DW interview.”

2) In the subsection “Recruitment and trial procedures” we added information about the physical location of the baseline evaluation: “A flyer, which included a general overview of the study and contact details, was disseminated via e-mail to the MS Centre patients by the MS Centre team. People who showed interest in participating in the study were contacted by the study coordinator. Subsequently, one trained clinical psychologist not involved with the intervention and blinded to patient’s assignment (examining clinician) made an appointment with those patients who met the inclusion criteria and agreed to participate in the study, checked all eligibility criteria and performed the baseline evaluation (T0) in a dedicated room of the hospital.”

3) In the subsection “Qualitative study participants and recruitment” we reported information on the interviews. In order to be more precise, the subsection was modified as follows: “Participants were informed about the possibility of participating in an individual interview when recruited. They were invited to participate in the qualitative study by e-mail or telephone within three months of completing READY. They were then fully informed of the aims and requirements of this action and provided informed consent. All trial participants who received READY were invited to participate. The examining clinician conducted the interviews in a dedicated room of the hospital or by phone (when a participant was unable to visit the hospital).”

4) Details on the duration and the setting (in person or via phone) of the interviews were presented in the “Nested qualitative study results” subsection (see edits in red): “Of the total 32 participants who participated in a READY group, 30 accepted to be interviewed (18 READY, 12 relaxation + READY; mean duration of interviews = 33.5 minutes, range 16 – 47; 25 in person).”

Could the authors clarify why they used computer-based stratified randomization (2 factors: Expanded Disability Status Scale (EDSS) [42] score < 2.0 and ≥ 2.0, and CDRISC score < 50 and ≥ 50).

In view of the limited size of the study, we used stratified randomization to prevent between-arm unevenness in important patient characteristics. Thus, we identified these characteristics: level of impairment/disability (EDSS score) and resilience (CD-RISC 25 score). 

Sample size – In the discussion the authors suggest the small sample size may have led to null quantitative results. The methods state the sample size calculation was based on achieving a large effect size, as reported in other studies. Can the authors provide some further discussion here? If their sample size calculation was correct, then the small sample size argument may be less acceptable.

When the study was designed there was no published information on the effects of READY in a RCT. In accord with your comments and those of reviewer 1, we modified the sample size calculation section as follows (see edits in red): “A sample size of 32 patients (18 per arm) achieves 80% power to detect a difference of 14 (standard deviation 22) in the MSQOL-54 MHC in a design with three repeated measurements, assuming an intra-subject correlation between observations of 0.75, an alpha error of 0.05, and 20% of the patients lost to follow-up (S5 Appendix). Because no data on the READY for MS were available when the study was designed, our estimate was based on the large effect size for QoL (d = 0.80) assessed by the Profile of Health-related Quality of Life in Chronic Disorders scale in a RCT of a group mindfulness intervention for PwMS [58], and on available data on the MSQOL-54 MHC [49, 59]”.

This is a pilot study, and one of its purposes was to collect data that allow us to perform an adequate power calculation on the interventions used (READY and relaxation). See also “Conclusion” (see edits in red): “The READY program was well accepted by MS patients with varied socio-demographic and clinical characteristics, suggesting it has high utility and acceptability in an Italian clinical setting. Statistical analyses showed that READY was not more efficacious than relaxation. In contrast, qualitative data indicated that READY was viewed by participants as superior to relaxation; a finding that converged with four non-significant statistical trends supporting the efficacy of READY. Consistent with the ACT psychological flexibility framework, the qualitative data indicated that participants’ perceived improvements in resilience and QoL were due to the acquisition of skills related to the six core ACT processes. 

The Steering Committee and an international expert panel (four persons) jointly discussed the study findings in two dedicated meetings (13 and 24 January 2020) and outlined the multi-centre RCT evaluation of the Italian READY for MS program. It was a structured discussion, using a “PICO” (Population, Intervention, Comparator, Outcomes) format [77]. The eligibility criteria of the pilot RCT, as well as the READY program were deemed adequate by the panel (Population and Intervention). To limit possible involuntary contaminations, in the multi-centre RCT the control group will be run by a psychologist not involved in “READY for MS training program” and with no expertise in ACT or mindfulness interventions (Comparator). Following Chmitorz et al. [78] and based on the trend differences in the CD-RISC 25 [52] observed in the pilot RCT, change in resilience at three month follow-up will be the primary trial endpoint. In addition, a longer follow-up assessment scheduled six month post-intervention was decided to evaluate effect maintenance (Outcomes).”

The differences between qualitative and quantitative outcomes - could the authors offer some thoughts about this?

Thank you for this interesting comment. We enriched the discussion (edits are reported in red in the manuscript) and added the following references to support our arguments:

• Brannen J. Mixing Methods: qualitative and quantitative research. Aldershot, Ashgate; 1992.

• Bryman A. Quantity and Quality in Social Research. London, Routledge; 1995.

• Cox K. Assessing the quality of life of patients in phase I and II anti-cancer drug trials: interviews versus questionnaires. Soc Sci Med. 2003 Mar;56(5):921-34. DOI: 10.1016/s0277-9536(02)00100-4

• Moffatt S, White M, Mackintosh J, Howel D. Using quantitative and qualitative data in health services research -what happens when mixed method findings conflict? BMC Health Serv Res. 2006:8;6:28.

6. PLOS authors have the option to publish the peer review history of their article (what does this mean?https://journals.plos.org/plosone/s/editorial-and-peer-review-process#loc-peer-review-history). If published, this will include your full peer review and any attached files.

Thanks for the opportunity, we do want to publish the review history of this article.

---

## [Decision Letter · Decision Letter 1]

24 Mar 2020

A resilience group training program for people with multiple sclerosis: results of a pilot single-blind randomized controlled trial and nested qualitative study

PONE-D-19-32115R1

Dear Dr. Giovannetti,

We are pleased to inform you that your manuscript has been judged scientifically suitable for publication and will be formally accepted for publication once it complies with all outstanding technical requirements.

With kind regards,

Andrea Martinuzzi

Academic Editor

PLOS ONE

Additional Editor Comments (optional):

Reviewers' comments:

Reviewer's Responses to Questions

**Comments to the Author**

1. If the authors have adequately addressed your comments raised in a previous round of review and you feel that this manuscript is now acceptable for publication, you may indicate that here to bypass the “Comments to the Author” section, enter your conflict of interest statement in the “Confidential to Editor” section, and submit your "Accept" recommendation.

Reviewer #1: All comments have been addressed

Reviewer #2: All comments have been addressed

2. Is the manuscript technically sound, and do the data support the conclusions?

Reviewer #1: (No Response)

Reviewer #2: Yes

3. Has the statistical analysis been performed appropriately and rigorously? 

Reviewer #1: (No Response)

Reviewer #2: Yes

4. Have the authors made all data underlying the findings in their manuscript fully available?

Reviewer #1: (No Response)

Reviewer #2: Yes

5. Is the manuscript presented in an intelligible fashion and written in standard English?

Reviewer #1: (No Response)

Reviewer #2: Yes

6. Review Comments to the Author

Reviewer #1: (No Response)

Reviewer #2: All my comments have been adequately addressed and I am now satisfied with the paper as is. Pleas

7. PLOS authors have the option to publish the peer review history of their article (what does this mean?). If published, this will include your full peer review and any attached files.

Reviewer #1: No

Reviewer #2: Yes: Lambros Messinis

---

## [Editor Report · Acceptance letter]

26 Mar 2020

PONE-D-19-32115R1 

A resilience group training program for people with multiple sclerosis: results of a pilot single-blind randomized controlled trial and nested qualitative study 

Dear Dr. Giovannetti:

I am pleased to inform you that your manuscript has been deemed suitable for publication in PLOS ONE. Congratulations! Your manuscript is now with our production department. 

With kind regards,

on behalf of

Dr. Andrea Martinuzzi 

Academic Editor

PLOS ONE